# A Framework for Enhancing Stock Investment Performance by Predicting Important Trading Points with Return-Adaptive Piecewise Linear Representation and Batch Attention Multi-Scale Convolutional Recurrent Neural Network

**DOI:** 10.3390/e25111500

**Published:** 2023-10-30

**Authors:** Yu Lin, Ben Liu

**Affiliations:** 1Joint Lab of Data Science and Business Intelligence, Southwestern University of Finance and Economics, Chengdu 610074, China; 2School of Statistics, Southwestern University of Finance and Economics, Chengdu 610074, China

**Keywords:** stock market, important trading points (ITPs), Return-Adaptive Piecewise Linear Representation (RA-PLR), Batch Attention Multi-Scale Convolution Recurrent Neural Network (Batch-MCRNN), trading strategy

## Abstract

Efficient stock status analysis and forecasting are important for stock market participants to be able to improve returns and reduce associated risks. However, stock market data are replete with noise and randomness, rendering the task of attaining precise price predictions arduous. Moreover, the lagging phenomenon of price prediction makes it hard for the corresponding trading strategy to capture the turning points, resulting in lower investment returns. To address this issue, we propose a framework for Important Trading Point (ITP) prediction based on Return-Adaptive Piecewise Linear Representation (RA-PLR) and a Batch Attention Multi-Scale Convolution Recurrent Neural Network (Batch-MCRNN) with the starting point of improving stock investment returns. Firstly, a novel RA-PLR method is adopted to detect historical ITPs in the stock market. Then, we apply the Batch-MCRNN model to integrate the information of the data across space, time, and sample dimensions for predicting future ITPs. Finally, we design a trading strategy that combines the Relative Strength Index (RSI) and the Double Check (DC) method to match ITP predictions. We conducted a comprehensive and systematic comparison with several state-of-the-art benchmark models on real-world datasets regarding prediction accuracy, risk, return, and other indicators. Our proposed method significantly outperformed the comparative methods on all indicators and has significant reference value for stock investment.

## 1. Introduction

The stock market is a lucrative investment avenue that has drawn the sustained interest of many investors and researchers [1]. The efficient market hypothesis (EMH) posits that stock prices are determined by all available information and relevant news [2]. Based on this hypothesis, researchers have developed several renowned stock market analysis and prediction theories, such as the Elliott Wave Theory [3] and Gann Theory, attempting to unveil the intrinsic link between stock price fluctuations and market changes. These theories have paved the way for subsequent stock market prediction research.

Conventional methods for predicting stock prices rely mainly on historical data and time-series analysis techniques [4]. However, these methods often fail to effectively capture the inherent patterns of stock price movements due to the complexity, randomness, and non-stationarity of stock market data. Artificial intelligence technology has advanced rapidly in recent years, and deep learning methods have achieved remarkable success in multiple fields, such as image recognition, speech analysis, and natural language processing. Many researchers have applied deep learning methods to the financial field, especially stock price or return prediction, and have obtained good results. Among them, the deep neural network (DNN) [5], recurrent neural network (RNN) [6,7,8,9,10,11,12,13], and convolution neural network (CNN) [14,15,16,17,18] methods are widely used to mine the complex patterns of stock price changes due to their powerful modeling ability for non-linear data and good adaptability for sequential data. In addition, there are some noteworthy methods. For instance, Zhang et al. improved the cell units of Long Short-Term Memory (LSTM) networks through frequency decomposition [19]; Feng et al. enhanced the robustness of the model through adversarial training [20]; Xu et al. improved the model’s anti-interference ability by combining generative models with variational inference [21]; and Zhang et al. enhanced the predictive performance by combining a transformer with attention networks [22]. These methods provide beneficial references and lessons for improving models’ structure to enhance their adaptability to financial data features.

Nevertheless, the direct application of stock price or profit forecasts to real-world stock market transactions does not result in superior returns that align with the models’ projected outcomes, leading to a ‘high model, low return’ scenario. By thoroughly examining the pertinent literature and empirical data, it is evident that the values of significant metrics, such as the Mean Squared Error (MSE) and the Mean Absolute Error (MAE), have exhibited a consistent decline as the depth of the investigation has increased. However, by comparing the predicted curves with the true curves, we found that the prediction results seem to be just a lagged version of the data for the previous trading day, and the models do not capture the trends and turning points of the stock price changes, as shown in Figure 1. This may be caused by distance-based regression loss functions when training models. This loss function makes models use the value of the nearest time step as the model output after multiple iterations to minimize the model loss. This causes models to fail to predict the turning point when the stock changes, resulting in erroneous judgments and operations and thus losses.

Hence, we need to reconsider the feasibility and necessity of finding a model that can precisely forecast stock prices in real time. Stock prices are affected by both internal factors, such as the company’s performance, and external factors, such as market sentiment, unforeseen events, and government policies. These factors introduce noise and randomness into the stock market data, making it challenging to accurately predict stock prices or returns, even with the advanced capabilities of deep neural networks in handling time-series data. Moreover, minor price fluctuations induced by noise are hard to discern and exploit, while high-return trading opportunities can lead to lower risk and higher returns. Such opportunities often emerge after a period of the accumulation of endogenous driving forces, which result in significant price movements. The optimal trading points are those close to the onset of these movements. Therefore, instead of pursuing accurate price prediction, which may be unattainable, it may be more worthwhile to investigate the endogenous patterns preceding optimal trading points and identify potential high-return trading opportunities.

Significant trading opportunities with high returns usually exist in the medium- and long-term trends of stock price changes rather than short-term fluctuations, which is consistent with the experience of traditional stock market investors. Due to their own limitations, human traders cannot traverse all stocks at any time to find short-term trading opportunities like machines. They use various analytical methods to judge the trend of stock prices and enter the market in advance, holding for a while. Their investment experience usually has the following three characteristics:**A focus on high-return trading opportunities:** Stock investors are not concerned about the price of every moment, but rather the important trading points that can bring high returns. The reason is that predicting stock prices is complex and unstable, and the high transaction cost makes small price fluctuations meaningless. Stock investors prefer trading points with clear signals to achieve higher returns. For example, a widely used trading strategy is Trading Range Breakout (TRB), which regards price breakout points as high-return trading opportunities. According to the experience of stock investors, entering the market when the stock price breaks through the trading range can bring higher returns. As shown in Figure 2, a breakout occurs when the stock price rises with an increased trading volume, exceeding the set resistance level. At this time, the stock price will break through the price barrier, volatility will increase, and the price will move quickly toward the breakout, forming a high-return trading opportunity favorable for long positions. At the same time, the careful observation of the stock price changes before the range breakout can reveal that before the price breakout, the stock price had already begun to turn and had accumulated momentum for a while. During this time, the changes in stock price and trading volume often hide the inherent change rules before the price breakout. If we can mine and grasp this rule, we can establish long positions in the price range before the breakout and achieve high-return and low-risk trading.**Grasping the changing patterns:** The stock price signal at a single moment is often insufficient, and many lucrative trading opportunities emerge when the stock price trend shifts significantly, which necessitates sustained observation for a while to ascertain the stock price movement. Hence, stock investors typically make an integrated assessment based on the trading price, trading volume, and other data over an interval to forecast the stock price’s future trajectory more precisely. Figure 3 illustrates four prevalent candlestick charts of stock price reversals. It can be observed that these classic candlestick formations are constituted by a sequence of historical data rather than an isolated instant. These data, over time, constitute a coherent signal, revealing the psychological and behavioral alterations of market participants during that span, thereby offering vital clues for the subsequent direction of stock prices.**Diversifying the investment portfolio:** The stock market occasionally undergoes a bull market scenario, i.e., the stock price persistently ascends, instilling confidence in investors. Nevertheless, the market trend is not invariably smooth sailing and may invert or plummet unpredictably. Consequently, prudent investors always abide by the "diversifying risk" strategy and allocate their funds across different domains and sectors. The advantage of this is that a diversified portfolio can augment returns and diminish risk levels [1,23]. Simultaneously, diversifying investment can also fortify investors’ capacity to handle market fluctuations and unforeseen events and preclude suffering substantial losses due to excessive reliance on one or a few stocks.

To achieve high returns with a certain level of risk by accurately predicting ITPs, three key issues must be addressed: the detection of historical ITPs, the prediction of future ITPs, and the construction of trading strategies. Combined with the practical experience of stock investors, we designed the following solutions.

First, for the detection of historical ITPs, we follow the basic principle that stock investors focus on high-return trading opportunities and use a novel Return-Adaptive Piecewise Linear Representation (RA-PLR) technique to define ITPs as turning points at which stock prices rise or fall sharply. This allows for a trading strategy that not only accurately captures buying opportunities, but also avoids declining returns or even losses due to missed selling points.

Second, for future ITP prediction, by drawing upon the insights gained from researchers who have explored various model structures and methods in financial markets and integrating them with stock investors’ interest in trend reversal patterns, we propose a Batch Attention Multi-Scale Convolution Recurrent Neural Network (Batch-MCRNN). This network combines the advantages of multi-scale convolution networks, stacked LSTM, and time and sample attention mechanisms to explore the potential rules of stock price fluctuations from three dimensions: space, time, and samples. It improves the prediction accuracy of future ITPs. Specifically, we use a Multi-Scale One-Dimension Convolution Neural Network (MS-1DCN) to convolve stock price sequences at different time scales in order to capture the local fluctuation characteristics of stock price changes and help the model focus on local persistent turning patterns. Then, we use Long Short-Term Memory (LSTM) and the temporal attention mechanism, which can perform effective long-sequence learning, to learn the long-term temporal dependency of the raw data and the local features obtained by the CNN. Finally, through the batch attention mechanism for sample interaction, the model itself can explore information between samples and achieve multi-dimensional information mining. In addition, we also introduce residual connections and batch normalization techniques to enhance the stability and generalization ability of the model.

Third, for the construction of trading strategies, the diversification of investment portfolios has been proven effective. We use the CSI 300 Index Constituents, which can reflect the overall situation of China’s stock market and contain numerous mainstream industries, to establish a dataset and achieve overall data diversification. On the one hand, historical data for stocks of the same type are not sufficient to train a robust neural network, which may lead to the overfitting of the model. Using richer data for training can allow neural networks to learn more generalized stock fluctuation rules. On the other hand, the strict definition of ITPs results in their scarcity, leading to an imbalance in the overall dataset. Using more stock data can facilitate subsequent sampling and help reduce the impact of data imbalance. Finally, based on a trading strategy compatible with the overall model, we design a Double Check (DC) method to filter out low-confidence and high-risk predictions and improve the overall performance of trading strategies.

Based on the analysis conducted previously, we propose a framework for ITP prediction based on RA-PLR and Batch-MCRNN with the starting point of improving stock investment returns. The framework aims to accurately define and predict ITPs to identify profitable trading opportunities in the stock market. We evaluated the performance of our method through rigorous and systematic experiments on real-world stock data and contrasted it with several state-of-the-art benchmark methods. The experimental results demonstrated that our method outperformed the baselines in prediction accuracy, return rate, and risk control. Therefore, we contend that it is more advantageous to focus on ITPs rather than the stock price or trend at every time point when developing a trading strategy. The main contributions of this work are summarized as follows:We propose a novel Return-Adaptive Piecewise Linear Representation for detecting historical ITPs. The method trades off the relationship between the number of ITPs and benefits from a profit-maximization perspective in order to extract reasonably significant trading points that truly satisfy investors’ expectations.We propose a Batch Attention Multi-Scale Convolution Recurrent Neural Network to forecast future ITPs. This network integrates the benefits of CNN, LSTM, and attention mechanism structures and enhances the prediction accuracy of ITPs.We propose an adaptive trading strategy that executes buy and sell operations based on ITPs. In addition, to enhance the strategy’s performance and reduce the impact of model prediction errors on the returns, we propose a Double Check method that improves the overall returns of the trading strategy.We conduct comprehensive and systematic experiments on real-world datasets. The effectiveness and robustness of the method proposed in this work are demonstrated in various aspects, from model prediction performance to simulated trading returns.

The rest of the work is organized as follows: Section 2 proposes and details a new methodology for improving stock investment returns based on the results of the previous analysis; Section 3 introduces and discusses the experimental settings and results to demonstrate the comparative advantages of the proposed method; Section 4 introduces the market simulation situation and analyzes the profitability and risk control capabilities of all methods; and, finally, Section 5 concludes the work and suggests future research directions.

## 2. Methodology

Based on the preceding section’s analysis and discussion, we propose a framework for ITP prediction based on RA-PLR and Batch-MCRNN. The framework comprises four primary components: data preprocessing, ITP detection, model training and ITP prediction, and trading strategy and return appraisal. The general procedure of the framework is illustrated in Figure 4. This part will focus on the components of the framework related to the detection of ITPs, the training of the model, and the prediction of ITPs. The data preparation technique will be discussed in the experimental portion of Section 3, while the trading strategy and return evaluation component will be examined in Section 4.

This work proposes a return evaluation function aligned with the trading strategy’s primary objective: enhancing return. This function aims to achieve effective ITP detection by trading off the number of trading points against the return size. It allows for filtering out high-return trading points that conform to investors’ expectations. In addition, we improve the detection of ITPs to mitigate the problems of sample imbalance and non-turning ITPs. For ITP prediction, this work develops a Batch Attention Multi-Scale Convolution Recurrent Neural Network. This network consists of a multi-scale one-dimension convolution network, a stacked recurrent neural network, and a time and sample attention mechanism, which delve into the latent patterns of stock data in the space, time, and sample dimensions. The multi-scale one-dimension convolution network can detect the fluctuation features of stock data at different scales; the stacked recurrent neural network can model the temporal dependency of stock data; and the time and sample attention mechanism can adaptively distribute the importance of different samples, time steps, and features. The method proposed in this work is elaborated in detail in the following sections.

### 2.1. Technical Factors

The prediction of the stock market encompasses a consideration of multiple input variables, including historical price data (such as opening, closing, high, and low prices) and trade volume. Nevertheless, it is essential to note that these fundamental characteristics fail to sufficiently encompass the intricate nature and ever-changing dynamics of the market. As a result, some researchers have added technical factors as critical features for their models [24]. These factors are computed from price and volume data and can effectively indicate the market’s state and behavior regarding trends, volatility, strength, momentum, etc. [25]. We selected several technical factors to supplement the essential data and improve its quality and information content based on domain experts’ experience and the relevant literature. Table 1 displays some of the primary technical factors that we employ.

### 2.2. Return-Adaptive ITP Detection

The successful prediction of future ITPs and the attainment of reliable profits depend on the effective identification of ITPs in historical stock price time series. This work employs PLR to define ITPs as peaks, troughs, and turning points based on an analysis of stock market investors’ behavior, a literature review, and the fundamental objective of trading strategies. Furthermore, we propose a return-adaptive function to trade off the number of trading points against the size of the return. This allows for filtering out high-return trading points that conform to investors’ expectations. Moreover, this work improves the ITP detection process by resolving the issues of sample imbalance and non-turning ITPs. The specific components are as follows.

#### 2.2.1. Piecewise Linear Representation (PLR)

PLR was first developed as a time-series segmentation linear approximation technique for pattern matching and is one of the most commonly used time-series segmentation representations. In financial time-series analysis, researchers often use it to generate turning points [26], which usually reflect the market’s trading signals. Suppose that $T=\left\{x_{1},x_{2},\ldots ,x_{l}\right\}$ represents financial time-series data, which can be decomposed into *K* segments by approximating lines. The segmented description is as follows:(1)$$\begin{array}{ccc} T_{PLR}= & \left\{L_{1}\left(x_{1},\ldots ,x_{t_{1}}\right),L_{2}\left(x_{t_{1}+1},x_{t_{1}+2},\ldots ,x_{t_{2}}\right),\ldots ,\right.\; & \left.L_{k}\left(x_{t_{k-1}+1},x_{t_{k-1}+2},\ldots ,x_{l}\right)\right\}, \end{array}$$
where $t_{i}$ is the end time of the *i*th segment, and $L_{i}\left(x_{t_{i-1}+1},x_{t_{i-1}+2},\ldots ,x_{t_{i}}\right)\left(1\leq i\leq k\right)$ denotes the approximate linear function of $x_{t_{i-1}+1},x_{t_{i-1}+2},\ldots ,x_{t_{i}}$. Since $t_{i}$ indicates the moment when the trend of the time series changes, it is often referred to as a turning point.

#### 2.2.2. Return-Adaptive Threshold δ Selection

In this work, the top-down PLR algorithm is used to segment stock time-series data for linear approximation in order to minimize the number of segments while controlling the segmentation error so that it does not exceed a preset threshold. In PLR segmentation, the threshold value δ is a key parameter affecting the results, and its size is inversely related to the number of segments. Therefore, choosing a reasonable δ threshold is essential for identifying turning points and making accurate predictions. Table 2 shows the number of turning points generated by PLR with different threshold values of δ.

In order to achieve accurate ITP detection, a suitable threshold δ needs to be chosen. The primary objective of ITP detection is to enhance returns. Hence, this study develops a return-adaptive threshold selection function from the standpoint of returns, considering the actual characteristics of stock market prediction and the identification of ITPs. The function is return-oriented, focusing on medium- and long-term trend changes rather than short-term volatility adjustment, which is consistent with the goal of this work to seek high-return trading opportunities under medium- and long-term trends. The specific form of the function is as follows:(2)$$R_{PLR}=Return_{PLR}-\alpha_{PLR}\sum_{i=2}^{n}\max\left(\beta_{PLR}-\left(x_{i}-x_{i-1}\right),0\right),$$
where $Return_{PLR}$ is the cumulative return corresponding to the turning point obtained based on the PLR algorithm, $\alpha_{PLR}$ is the penalization factor, and $\beta_{PLR}$ is the expected time interval of the trade, which is penalized if the time interval between the two turning points is less than this value. The degree of penalty is inversely proportional to the time interval. $x_{i}$ is the date of the *i*th turning point. Since the data are known during the detection phase, it is possible to trade according to the principle of buying low and selling high. Obviously, the denser the turning points, the higher the cumulative return, i.e., the larger the $Return_{PLR}$. However, at the same time, overly frequent trading will be affected by the penalty term. Therefore, the parameter δ of the PLR algorithm can be chosen automatically by maximizing the objective function $R_{PLR}$, so that the cumulative return is balanced with the number of turning points. Thus, the turning points are converted into ITPs that meet the investors’ expectations.

#### 2.2.3. Non-Turning ITP Optimization

Since the segmentation mechanism of PLR does not require two turning points to have opposite trend directions, lowering the threshold increases the number of ITPs, resulting in some short-term invalid turning points for PLR. These points only reflect short-term price fluctuations rather than medium- or long-term trend changes. Figure 5 illustrates an example. In Figure 5a, several groups of ITPs with the same trend appear in a short time span. The ITPs marked with green solid boxes actually belong to the same downward or upward trend and should not be considered as ITPs. Therefore, after removing the short-term fluctuation points, Figure 5b shows a more reasonable selection of ITPs.

#### 2.2.4. Data Augmentation

This work focuses on the medium- and long-term trends of stock price changes and seeks high-return trading points rather than short-term fluctuations. Therefore, the ITPs focused on in this work pertain to a low proportion of the overall data, less than 10%, leading to a severe data imbalance. Data imbalance affects the learning performance and evaluation accuracy of deep learning methods. A standard solution is to balance the number of samples in each category by undersampling or oversampling. However, undersampling loses much helpful information, while oversampling increases the risk of overfitting. To overcome these problems, we find that the neighborhood features of ITPs are very close to the ITPs themselves, making it difficult to distinguish which neighborhood points are ITPs. Therefore, we combine the experience of domain experts and define ITPs as an interval rather than a point. In this case, the neighborhood points of ITPs generated by the PLR are also labeled as ITPs. Nevertheless, determining the size of the ITP interval is a new problem. The simplest method is to use a fixed-length window. However, as seen in Figure 6a, there are differences in the change rate within each ITP neighborhood. When the change rate is large, the stock is in a violent fluctuation state, and the neighborhood points should be as close as possible to the original ITPs to maintain the relative consistency of the ITP features. Conversely, the stock price is stable when the change rate is small, and the number of neighborhood points can be slightly increased. In summary, when the change rate of one side of the neighborhood is greater than the threshold τ, the sampling window size is $N_{low}$; when the change rate is less than the threshold τ, the sampling window size is $N_{high}$. Figure 6b shows the data points after enhancement.

### 2.3. Batch Attention Multi-Scale Convolution Recurrent Neural Network (Batch-MCRNN)

This work proposes a Batch Attention Multi-Scale Convolution Recurrent Neural Network (Batch-MCRNN) model based on the characteristics of stock time series, stock investors’ attention to local turning patterns, and the applicability of deep learning methods in finance. The model comprehensively mines the intrinsic laws of stock data from three dimensions—space, time, and samples—to improve the accuracy of ITP prediction and provide support for improving the returns of trading strategies. The model is mainly composed of the following parts: a multi-scale one-dimension convolution layer for processing local features; a fusion embedding layer for fusing features; a stacked LSTM layer for learning temporal features; an attention and residual connection layer for enhancing sequential attention; and a batch attention layer for reinforcing sample interaction. The overall structure of the model is shown in Figure 7. Each component is described in detail below.

#### 2.3.1. Multi-Scale One-Dimension Convolution Layer

Recent years have witnessed rapid advances in artificial intelligence technology, and CNNs, as an effective method for feature extraction, have shown outstanding performance in various fields besides computer vision. In particular, one-dimension CNNs excel at handling time-series data [27]. This paper employs an MS-1DCN to capture local fluctuation features in stock sequence data. By applying convolution kernels with different scales, a model can recognize local features across different scopes, thus improving its expressiveness. Moreover, this paper also utilizes cross-channel convolution kernels to enhance the information communication among various features. Figure 8 illustrates that each one-dimension convolution layer under each convolution scale consists of several same-sized convolution kernels. These convolution kernels move along the time axis with a fixed stride to extract more abstract and representative local features from the stock time series. The extracted features are padded to generate new sequence features, which act as the input for the subsequent convolution layer.

Given a multidimensional stock historical data time series $\left\{X_{t}|t=12,\ldots ,L\right\}$ for a single dimensional sequence $\left\{x_{t}|t=1,2,\ldots ,L\right\}$ for each input time step $x_{t}$, the following equations show the result of a single convolutional kernel at a single time step under a single channel:(3)$$x_{j}^{l}=f\left(\sum_{i\in M_{j}}x_{i}^{l-1}*k_{ij}^{l}+b_{j}^{l}\right),$$
where $M_{j}$ denotes the selection of the input mapping; $x_{j}^{l}$ represents the result of the *j*th feature mapping in the *l* layer, i.e., the output of the convolution; $x_{i}^{l-1}$ represents the *i*th feature mapping in the l−1 layer, i.e., the input of the convolutional layer; $k_{ij}^{l}$ represents the weights of the *i*th feature mapping in the *l* layer; $b_{j}^{l}$ represents the bias vectors of the convolutional layer; and f(-) is the activation function. In order to perform feature fusion embedding between the convolved local features and the original sequence, we padded the input sequence with 0 of length 2p−1 (*p* is the convolution kernel size) to keep the sequence length constant during convolution. In the subsequent analysis, we will use ${\mathrm{Conv}}_{ab}$ to denote the convolutional output of the *b*th channel at the *a*th scale.

#### 2.3.2. Fusion Embedding Layer

The MS-1DCN consists of S one-dimension convolution layers with different scales, each with M channels, and outputs of S*M new feature sequences reflecting the local fluctuation information at different scales. Then, these new feature sequences are spliced with the original stock sequences to obtain an enhanced feature sequence. Finally, the enhanced feature sequence is converted into input data for LSTM by an embedding network with the following equation:(4)$$\bar{X}=\mathrm{Embeding}\left(\mathrm{Concat}\left(\left[{\mathrm{Conv}}_{ab},X\right]\right)\right),$$
where a=1,2,…,S, b=1,2,…,M. The advantage of the fusion embedding is that the global information in the original stock series is retained while new local volatility features are extracted, thus improving the accuracy and robustness of the prediction.

#### 2.3.3. Stacked LSTM Layer

After the fusion embedding layer, the local features are combined with the original data to aggregate a new vector, and this joint feature is fed into the LSTM layer to learn the long-term temporal dependencies. We employ a multi-layer stacked LSTM structure, which can accelerates the convergence process. To prevent overfitting and improve the generalization ability of the model, we add dropout and batch normalization operations after each LSTM layer. The two-layer stacked LSTM model can be briefly described as follows:(5)$$\begin{array}{cc} \left(h_{t}^{1},c_{t}^{1}\right) & =\mathrm{LSTM}1\left(\bar{X_{t}},h_{t-1}^{1},c_{t-1}^{1},W^{1},b^{1}\right)\\ \left(h_{t}^{2},c_{t}^{2}\right) & =\mathrm{LSTM}2\left(h_{t}^{1},h_{t-1}^{2},c_{t-1}^{2},W^{2},b^{2}\right). \end{array}$$

#### 2.3.4. Attention and Residual Connection Layer

In this work, we use the temporal attention mechanism, which is a variant of the standard attention mechanism for sequential data. Temporal attention is a self-attention mechanism that eliminates the need for an extra decoder. Instead, it calculates the alignment scores at each time step *t* of the encoder, using the weighted sum of the encoder’s hidden states at the current and other time steps. The formula is as follows:(6)$${\tilde{\alpha }}_{t}=u_{a}^{T}\tanh\left(W_{a}h_{t}+b_{a}\right),$$
where $W_{a}$ and $u_{a}$ are trainable weights called attention weights. The weights $W_{a}$ and $u_{a}$ are associated with the hidden states of the encoder. For each time step *t*, the attention weights $\alpha_{t}$ are obtained by normalizing the score ${\tilde{\alpha }}_{t}$ over all time steps of the encoder using the softmax function:(7)$$\alpha_{t}=\frac{\exp({\tilde{\alpha }}_{t})}{\sum_{t=1}^{T}\exp\left({\tilde{\alpha }}_{t}\right)}.$$

The importance of the output at time step *t* is indicated by the attention weight $\alpha_{t}$. The context vector *a* is a weighted sum of all hidden values of the encoder, where the weights are determined by the attention weights:(8)$$a=\sum_{t=1}^{T}\alpha_{t}h_{t},$$
where $W_{a}\in R^{E^{\prime }\times U}$, $b_{a}$, and $u_{a}\in R^{E^{\prime }}$ are the parameters to be learned, and *a* is the weighted representation of the complete set of hidden states in the sequence.

To improve the convergence speed and performance of CNN networks, researchers have designed a deep residual network (ResNet) utilizing residual connection, the basic principle of which is to define the objective function of the residual connection through Equation (Equation 9):(9)H(x)=F(x)+x,
where *x* is the input, F(X) is the residual function, and H(X) is the objective function. Instead of fitting the objective function H(X) directly, each residual cell fits the residual function F(X)=H(X)−x. Based on this idea, we add residual connection after the output of the attention mechanism, which enables the model to learn more effective residual functions and improves the expressive ability of the model. In this work, we connect the attention feature vector *a* with the final hidden state $h_{T}^{n}$ of the stacked LSTM network as a comprehensive representation of the time series, as shown below:(10)$$e={\left[a^{T},{\left(h_{T}^{n}\right)}^{T}\right]}^{T}.$$

#### 2.3.5. Batch Attention Layer

Inspired by the success of attention mechanisms in various tasks, we propose a batch attention module that enables deep neural networks to learn the relationships among samples within each batch automatically. In this way, the proposed method can achieve collaborative learning among samples. Figure 9 illustrates the learning framework using the batch attention module. Specifically, a backbone network is first used to learn representations for individual data samples without considering the interactions among different samples within each batch. Then, a batch attention module is introduced, which utilizes the multi-head attention mechanism from the transformer to model the relationships among different samples. The batch attention module’s output serves as the final classifier’s input. To reduce the gap between training and testing, an auxiliary classifier is used before the batch attention module, which shares weights with the final classifier and provides feedback to the batch attention module.

To model relationships between different samples, the batch attention module employs multi-head self-attention (MSA) and MLP blocks of the transformer structure and applies Layernorm(LN) for normalization after each block. Let the input feature sequence be $e\in R^{L\times C}$, where *L* is the length of the attentional residual link output sequence, and *C* is the dimensionality of the input features. The output of the transformer structure can be expressed as:(11)$$\begin{array}{cc} \; & {\hat{e}}_{l}=\mathrm{LN}\left(\mathrm{MSA}\left(e_{l-1}\right)+e_{l-1}\right)\\ \; & e_{l}=\mathrm{LN}\left(\mathrm{MLP}\left({\hat{e}}_{l}\right)+{\hat{e}}_{l}\right), \end{array}$$
where *l* is the layer index of the transformer’s multi-head self-attention layer. The multi-head attention layer effectively captures relational information [28] from the channel and spatial dimensions. Therefore, we believe it can also be extended to the batch dimension to explore interactions between different samples. Unlike the regular application of the transformer, the batch-focused input is first subjected to a reshaping operation that enables the transformer to apply the self-attention mechanism to the batch dimension of the input data. In this way, the self-attention mechanism becomes a cross-following mechanism across samples.

**Weight-sharing classifier.** The features before and after the batch attention module may be inconsistent due to the uncertainty of the batch size in the testing phase. This prevents us from simply discarding the batch attention module to infer new samples. To solve this problem, we introduce a new auxiliary classifier that can learn knowledge from the final classifier as well as keep consistent with the features before the batch attention module. Specifically, we only have to share weight parameters among the auxiliary and final classifiers. We term this simple and effective approach the “weight-sharing classifier”. With the “weight-sharing classifier”, we can eliminate the batch attention module in the testing phase while still utilizing the sample relation learning performed by the batch attention module. The auxiliary classifier provides the final output of the Batch-MCRNN model, as shown below:(12)$${\hat{y}}_{t}=We_{l}+b.$$

### 2.4. Returns Retrospective Threshold Search

Classifying ITPs is approached as a binary classification problem in our work. The model’s output is the probability assigned to each time point, indicating its likelihood of belonging to an ITP. It is necessary to establish a threshold θ to convert the probability output into classification labels:(13)$${\mathrm{TP}}_{t}={\hat{y}}_{t}\geq \theta .$$

The predicted label at time point *t* is denoted by ${\mathrm{ITP}}_{t}$. The prediction is an ITP (“1”) if $\hat{y}⩾\theta$; otherwise, it is a non-ITP (“0”). The threshold θ plays a crucial role in the model’s predictive performance. We can set θ to 0.5 when the dataset has balanced classes. However, our problem involves data imbalance, as the proportion of ITPs remains much lower than that of non-ITPs even after data augmentation. A simple and effective solution is to adjust θ according to the ratio of ITPs in the training set. Alternatively, we can optimize θ based on the precision–recall or ROC curves. However, these methods may not account for the temporal dependence of time-series data. Therefore, we introduce a returns retrospective threshold search mechanism. It does not target prediction accuracy but returns maximization. The optimal threshold is determined by backtesting the returns of different thresholds in the validation set period. The process is illustrated in Figure 10.

## 3. Research Design and Experimental Results Analysis

This section first describes the research design. Then, the performance of the proposed Batch-MCRNN model is evaluated in terms of its ITP prediction ability through comparative experiments on real-world datasets with several state-of-the-art benchmark methods.

### 3.1. Data Preprocessing

We acquired the trading data from 1 January 2014 to 31 December 2021 from the Tushare website. The dataset was composed of the constituents of the CSI 300 Index as of 31 December 2021. The CSI 300 Index is a composite index of the Shanghai and Shenzhen markets’ overall trends, capturing most of the market size and principal industries, with a high market representativeness and liquidity level. Its constituents are excellent active core investment stocks in various areas, revealing the return condition of the market’s dominant investments.

Since the trading suspension of listed companies interferes with stock prices and causes them to deviate from normal fluctuations, we excluded stocks whose trading days accounted for less than 98% of the total trading days in the experiment to eliminate the impact of outliers caused by frequent trading suspension. Moreover, stock prices that fluctuate at low levels for a long time tend to lack investment attractiveness and affect the model’s ability to identify ITPs. Therefore, we also excluded stocks whose low-level fluctuation ratio exceeded 50%. The specific screening criteria were as follows:Stocks whose non-trading suspension time accounted for no less than 98% of the total trading days;Stocks whose price was RMB 4 per share or less for no more than 50% of the entire duration.

Through the filtering process, we obtained 100 quality stocks. Then, we performed feature scaling on the original stock and technical factor data to eliminate the effects of scale differences among different stocks and enhance the convergence speed before feeding them into the neural network. We used the following formula for feature scaling without a loss of generality:(14)$$X_{\mathrm{norm}}=\frac{X-X_{\min}}{X_{\max}-X_{\min}},$$
where *X* is the original data; $X_{max}$ and $X_{min}$ are the maximum and minimum values before normalization, respectively; and $X_{norm}$ is the normalized dataset. To meet the requirements of time-series prediction, we divide the data into three parts according to the temporal order: the training set, the validation set, and the test set. The diagram of data division is shown in Figure 11.

### 3.2. Evaluation Metric

The rarity of ITPs and the stringent detection conditions result in an unbalanced dataset. This renders the accuracy meaningless for evaluating the performance of the classifier, as it cannot effectively identify the ITPs. Therefore, three evaluation metrics robust to unbalanced datasets were used in this work: AUC, F1 score, and MCC. Their definitions and computational formulas are given below.

(1) AUC. The AUC is the area under the ROC curve [29], which reflects the separability of the classifier. The value of the AUC is in the range of [0, +1]. An AUC of 0.5 indicates that the classifier cannot separate categories. The formula for the AUC is
(15)$$AUC=\int_{0}^{1}ITPR\left({FPR}^{-1}\left(x\right)\right)dx,$$
where TPR indicates the true-positive rate, and FPR indicates the false-positive rate.

(2) MCC. A holistic indicator of a classifier’s quality is the Matthews correlation coefficient (MCC), which synthesizes the information from the four components (ITP, TN, FP, FN) of the confusion matrix. It is stable regardless of the positive and negative sample imbalance across different classes [30]. The MCC has a value between −1 and +1, with values near +1 signifying an excellent classifier and near −1 signifying a poor classifier. A classifier with an MCC of 0 has no ability to separate classes. The MCC is computed as follows:(16)$$MCC=\frac{TP\cdot TN-FP\cdot FN}{\sqrt{\left(TP+FP)\cdot (ITP+FN)\cdot (TN+FP)\cdot (TN+FN\right)}},$$
where TP and TN are the numbers of correctly predicted positive and negative samples, and FP and FN are the number of incorrectly predicted positive and negative samples, respectively.

(3) F1 score. Precision and recall are combined into a single metric called the F1 score, which is the harmonic mean of the two. The F1 score can vary from 0 to 1, with higher values denoting better classifiers, and lower values indicating worse classifiers. The F1 score is more concerned with the predictive effectiveness of positive samples and is insensitive to whether the labels of positive and negative samples are balanced [31]. The F1 score is calculated using the following formula:(17)$$F1\;\mathrm{score}\;=\frac{2\cdot TP}{2\cdot TP+FP+FN}=2\cdot \frac{\;\mathrm{precision}\;\cdot \;\mathrm{recall}\;}{\;\mathrm{precision}\;+\;\mathrm{recall}\;}.$$

### 3.3. Compared Methods

This section aims to conduct a comparative analysis between the Batch-MCRNN model and several state-of-the-art benchmark models, while ensuring that all models are evaluated under identical conditions. The comparison models are briefly described below:

**(1) MCRNN:** MCRNN denotes the removal of the batch attention module in Batch-MCRNN, mapping the output of the residual connection to the labels directly through the fully connected layer and the sigmoid function.

**(2) NCNN:** NCNN denotes the removal of the MS-1DCN used for local feature extraction in Batch-MCRNN. The input data pass directly to the embedding layer and are fed into the subsequent network after passing through the embedding, and the remaining structure remains unchanged.

**(3) SFM:** Zhang et al. proposed a novel State Frequency Memory (SFM) recurrent network for capturing multi-frequency trading patterns from past market data and making long-term and short-term stock price forecasts based on these patterns [19].

**(4) AdvNet:** Feng et al. proposed a novel machine learning solution using adversarial training to improve the generalization of neural network predictive models for predicting whether stock prices will rise or fall in the future [20].

**(5) StockNet:** Xu et al. proposed a novel deep generative model that introduces recurring continuous hidden variables for dealing with the high degree of randomness in the market and uses neural variational inference to address the unsolvability of a posteriori inferences for predicting the rise and fall of stock prices [21].

**(6) TEANet:** Zhang et al. proposed a deep learning framework based on transformer and attention networks that utilizes historical text and stock price signals to capture the time-dependence of financial data and predict the rise and fall of stock prices [22].

### 3.4. Results Analysis

The purpose of this part is to analyze and discuss the prediction performance of the Batch-MCRNN model along with the other six benchmark models (MCRNN, NCNN, SFM, TEANet, StockNet, and AdvNet) on 100 high-quality stocks selected from the constituents of CSI 300. In order to maintain consistency with the trading strategies in the next section, only the prediction results on 18 stocks with more than one trade are reported in this section. In contrast, the results for the remaining stocks are omitted due to space limitations. Table 3, Table 4 and Table 5 show the prediction results for each metric (AUC, F1 score, and MCC).

As shown in the three tables, the Batch-MCRNN model outperformed the other benchmark models for 15 out of 18 stocks in terms of the AUC metric, with a winning rate of 83.33%. Its mean value was 10.0% higher than the second-best model, and its standard deviation was reduced by 5.7%. In terms of the F1 and MCC scores, the Batch-MCRNN model also won 14 times, with a winning rate of 77.7% for both metrics. Its mean values were 8.8% and 12.8% higher than the second-best scores, respectively. These results indicate that the Batch-MCRNN model significantly surpassed the benchmark models for all metrics, thanks to the advantages of combining convolutional neural networks, recurrent neural networks, attention mechanisms, and residual connections. The Batch-MCRNN model could effectively extract local features, temporal features, and sample interaction information from stock data in three dimensions: space, time, and samples. This enhanced the model’s ability to suppress noise and learn complex patterns and improved its prediction performance without relying on additional data.

**Discussion on the role of MS-1DCN.** Sequential information in time-series data is effectively learned by most existing methods using RNNs as the primary model. Still, they ignore the local fluctuation features in the data, which are also very important for stock market prediction. We present a novel Batch-MCRNN model that fuses RNNs and CNNs, employing CNNs to extract local fluctuation features at multiple scales and RNNs to learn long-term temporal dependencies, thus achieving the comprehensive modeling of time-series data. The experimental results revealed that the Batch-MCRNN model surpassed the NCNN model on three evaluation metrics, proving that the MS-1DCN could enhance the model’s predictive ability.

**Discussion on the role of batch attention.** Data imbalance can lead to biased predictions from deep neural networks. Some standard solutions are to adjust the data distribution through sampling techniques in order to make the sample size of each class more balanced. However, these methods can only address the data imbalance problem at the input level and do not change deep neural networks’ internal structure and mechanism. To address the problem with the model itself, we design a batch attention module that allows the deep neural network to interact and propagate information between batch samples so that each sample can contribute to the learning of all classes. By comparing the performance of the Batch-MCRNN model and the MCRNN model on the three evaluation metrics, the Batch-MCRNN model significantly outperformed the MCRNN model. The batch attention module plays a vital role in model prediction.

## 4. Trading Strategies and Returns Analysis

In this section, we simulate stock trading for 2021 and test the profitability and risk control of the methods based on cumulative return and risk assessment metrics. In the backtest, all methods were traded on a day-by-day basis, with the closing price of the day as the trading price. Meanwhile, the market liquidity was assumed to be sufficient, and the slippage effect was not considered. According to the actual situation of the Chinese market, only long positions were held, and no short positions were held. The total cost per transaction (including stamp duty, commission, etc.) was 0.2%. Details of the evaluation metrics, trading strategies, and returns analysis are provided below.

### 4.1. Returns Evaluation Metrics

We used three recognized indicators to assess the return–risk characteristics of the various approaches. The first metric was the annualized rate of return (ARR), which is the ratio of annual realizable profit to principal. The second metric was the Sharpe ratio (SR), which reflects the level of excess return of a portfolio relative to a risk-free asset and is calculated as follows:(18)$$\;\mathrm{SR}\;=\frac{R_{P}-R_{f}}{\sigma_{P}},$$
where $R_{P}$ is the return of the portfolio, $R_{f}$ is the risk-free rate (we took 3% as the valuation of $R_{f}$), and $\sigma_{P}$ is the volatility of the portfolio’s excess return. The higher the value of SR, the higher the excess return of the portfolio.

The third metric was maximum drawdown (MDD), which measures the maximum magnitude of loss experienced by a portfolio before it reaches a new high and is calculated by the formula
(19)$$\;\mathrm{MDD}\;=\frac{\;\mathrm{Trough}\;-\;\mathrm{Peak}\;}{\;\mathrm{Peak}\;}$$

### 4.2. Trading Strategies

ITPs can only indicate the critical moments of price changes and cannot determine their trends. Therefore, we introduced the Relative Strength Index (RSI) as an auxiliary indicator to help the trading strategy determine whether the ITPs were buy points (BPs) or sell points (SPs).

The Relative Strength Index (RSI) is a prevalent momentum indicator that gauges the speed and magnitude of changes in stock prices, as well as the level of overvaluation or undervaluation of stocks. The RSI can show the overbought or oversold status of stocks. The RSI values vary from 0 to 100. Usually, an RSI exceeding 70 implies that stocks are overbought, while an RSI falling below 30 implies that stocks are oversold [32]. We used the RSI to assess stock trends and trading directions. Due to the inherent predictive error in the model, both excessively high and low RSI thresholds could lead to fewer trading opportunities. Therefore, by analyzing stock data where the balance between upward and downward movements was relatively even, we observed a bimodal distribution of RSI values, with two peaks occurring around 60 and 40. Consequently, we optimized the RSI thresholds for trading direction at 60 and 40. The specific decision rules were as follows:(20)$$\left\{\begin{array}{c} \;\mathrm{If}\;RSI_{14}\left(t\right)<40,\;i.e.,\;t\;\mathrm{is}\;a\;\mathrm{buy}\;\mathrm{point}.\;\\ \;\mathrm{If}\;RSI_{14}\left(t\right)>60,\;i.e.,\;t\;\mathrm{is}\;a\;\mathrm{sell}\;\mathrm{point}.\; \end{array}\right.$$

After determining the ITPs’ trading direction, we designed a comprehensive trading strategy to assess the returns and risk management capabilities of each model. The specific trading strategy was as follows: We allocated a fixed amount to each of the 100 stocks in the CSI 300 and managed their positions separately. In 2021, we slid from the first trading day until we encountered the first buy point, entered a full position, and exited a full position when we encountered the first sell point. We disregarded any other buy signals during this period. If the position was still held at the end of the trading period, we liquidated it on the final day. Finally, we computed the individual metrics for each stock and the aggregate metrics for all stocks.

To mitigate the negative impact of model prediction errors, we propose an improved method based on the Delayed One-Day Strategy (DODS) [33] called the Double Check (DC) method. This method combines the rate of change of the stock’s prior return cr(t) and the RSI indicator to perform a secondary screening of the trading points predicted by the model. The core principle of the DC method is that if the stock price change trend moderates at the time of buying or strengthens at the time of selling, then the trade is made in accordance with the indication of the RSI indicator. Otherwise, the transaction is canceled. The specific steps of the DC method are shown in Algorithm 1, where the formula for the calculation of cr(t) is as follows:(21)$$cr\left(t\right)=\left(P_{c}\left(t\right)-P_{c}\left(t-1\right)\right)/P_{c}\left(t-1\right)$$
**Algorithm 1:**Double Check (DC) method
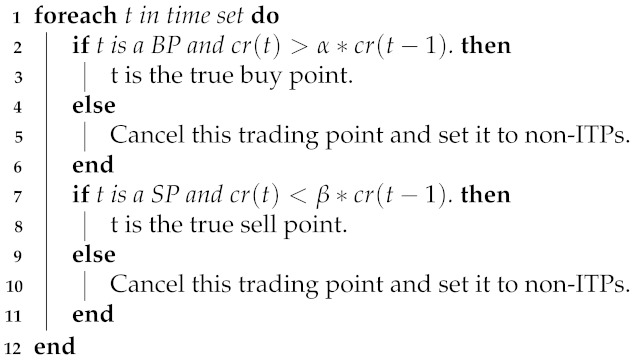


In Algorithm 1, α and β represent the smoothing coefficients. We employed a grid search approach to determine the optimal alpha and beta values that maximized returns in the validation set after threshold searching using a specific model and threshold. In our experiments, these values were both set to 0.9.

### 4.3. Returns Analysis

Figure 12 depicts the market backtesting outcomes of all models applied to 100 constituent stocks from the CSI 300 index over 243 trading days in 2021. Three more auxiliary evaluation indicators were introduced to enhance the objectivity of the performance evaluation: the winning rate, maximum individual stock returns, and maximum individual stock losses. This allowed us to analyze and evaluate the forecast results more comprehensively. For ease of presentation, we abbreviate AdvNet, StockNet, and TEANet as ANet, SNet, and TNet, respectively, in the figure.

Overall, the Batch-MCRNN model proposed in this work achieved an annualized return of 62.3%, a winning rate of 80.9%, and a maximum retracement of 16.4% in the framework of ITP prediction. Furthermore, the CSI 300 plunged more than 1000 points over the same period, or 21.3%, in a downward trend, demonstrating the effectiveness of Batch-MCRNN in improving the stock market investment return and reducing the investment risk.

For a trading strategy, there are two critical criteria for evaluation: the net profitability within a certain trading period and the risk control ability of the strategy.

**Discussion on profitability.** In the introduction, we discussed the methods of predicting stock returns or prices. Although the models seemed to have good predictive performance, they could not achieve a breakthrough in returns, resulting in the strange phenomenon of “high model, low return”. The core of this work was to improve stock investment returns and reduce risk rather than just pursuing the predictive accuracy of the model, although our proposed Batch-MCRNN model performed excellently. From Figure 12a, we can observe that even considering the trading costs and a long trading period of one year, the worst-performing model achieved a compound return of 34.3% and a win rate of 67%, confirming the feasibility and effectiveness of stock investment by mining the potential change patterns of important trading points.

**Discussion on risk control capacity.** In the simulation, our approach traded 100 individual types of stocks at the same time, which greatly diversified the trading risk, and made 89 total trades (buys and sells) over a 1-year trading cycle. The win rate (percentage of profitable trades) was more than 80%, implying that there was a high likelihood of a gain on every trade. Regarding the risk–return profile, the overall model had a maximum drawdown of 16.4% and an average Sharpe ratio of 2.913 (calculated only for the 18 stocks traded more than once), which were better than usual. The strategy demonstrated reasonable risk control. Due to the low number of trades and the maximum individual stock loss not exceeding 20%, we did not use additional take-profit and stop-loss strategies to avoid a decline in overall profitability. Overall, the Batch-MCRNN-based trading strategy demonstrated reasonable risk control.

To illustrate Batch-MCRNN’s real-world trading performance more clearly, we chose 18 out of 100 stocks that traded more than once (in line with the previous section’s experimental results). Figure 13 and Figure 14 show their specific trading times and prices (red stars mark buying points, and green stars mark selling points). All 18 stocks made profits, ranging from 2.4% to 137.4%. These stocks covered different market trends, such as rising, oscillating, and falling, validating the adaptability and stability of the forecasting framework proposed in this work for various market environments.

Moreover, we observed that for highly volatile stocks like 000596.SZ, 300015.SZ, and 600600.SH, our method had a high predictive power and return rate. These stocks underwent drastic price changes within the year, offering investors lucrative trading opportunities. Our method also effectively detected the start and end of upward trends, entering and exiting promptly, and achieving low-risk and high-return trades. However, our method had some limitations—mainly, its poor prediction performance for long-term declining or mildly fluctuating stocks such as 600009.SH, 600031.SH, and the latter half of 002304.SZ. This may be related to the prediction framework’s design and the trading strategy. Firstly, we did not categorize stocks by volatility and trendiness when identifying ITPs, which significantly affected the model’s performance. Secondly, short-selling is challenging and expensive due to China’s market specificity. Hence, our trading strategy only considered long positions, hampering its downward trend performance.

There were also some interesting results. First, the returns of the 18 stocks exhibited no discernible link with the forecasted outcomes derived from the model discussed in the preceding section. For instance, 002475.SZ performed worse than 600436.SH according to the model’s evaluation metrics, but its return was much higher. We speculate that this was because the model predicted far more ITPs than the actual trading points, and the trading strategy proposed in this work failed to optimally match these predicted points, resulting in some deviations in the actual trading situation. Moreover, since our trading strategy forced close positions at the end of the trading cycle, many stocks suffered losses in the forced closing, as shown in the figures for 600196.SH, 600079.SH, 600031.SH, and so on. The loss was more noticeable when the stock was in a downward trend. These problems prompted us to think about improving our trading strategy, and we will strive to design a more reasonable trading strategy in our future work.

**Discussion on the role of the Double Check (DC) method:** As shown in Table 6, we adopted the same evaluation system as the other models to validate the effectiveness of the DC method based on the Batch-MCRNN model. As expected, the DC method conducted a second screening of the trading points, reducing the trading points and thus decreasing the trading frequency. However, indicators such as the win rate, total return, and maximum drawdown all improved to varying degrees. Suffice it to say that using the DC method in trading strategies improved the prediction results of trading points and achieved better profitability.

## 5. Conclusions

This work focused on improving stock investment returns based on the perspective of important trading points (ITPs). We proposed an ITP prediction framework based on Return-Adaptive Piecewise Linear Representation (RA-PLR) and a Batch Attention Multi-Scale Convolution Recurrent Neural Network (Batch-MCRNN) for three core problems: the detection of historical ITPs, the prediction of future ITPs, and the construction of trading strategies. The framework integrated the experience of stock investors seeking high-return trading opportunities and the advantages of different deep learning structures in dealing with non-linear and non-stationary data, achieving higher prediction accuracy and profitability. The framework consists of four main parts: data preprocessing, ITP detection based on RA-PLR, ITP prediction based on Batch-MCRNN, and a trading strategy combining the Relative Strength Index (RSI) and Double Check (DC) methods. Unlike existing research, this framework always aims to increase trading returns and reduce trading risks, excluding some abnormally performing stocks from the data preprocessing stage; designing a return-adaptive function to balance returns and risks in the ITP detection stage; using the advantages of convolutional neural networks, recurrent neural networks, and attention mechanisms to deeply mine the intrinsic patterns of data from the spatial, temporal, and sample dimensions in the ITP prediction stage; and combining the RSI and DC methods to enhance performance in the trading strategy stage. Extensive experiments on real stock market data showed that the framework could effectively control risk while making sizable profits. In summary, due to the uncertainty and difficulty of stock price prediction tasks, as well as the high transaction costs that make minor price fluctuations irrelevant, focusing on predicting important trading points that are more likely to bring high-return opportunities rather than predicting stock prices or trends at every time point can achieve more profits. This work provides a new perspective on the stock market prediction field, which could help investors develop better trading systems.

The above analysis showed that the input data’s volatility and trend significantly impact the overall return. We plan to solve this problem in the future by either adding an additional discriminant model at the data layer and applying the ITP forecasting framework only to stocks with high volatility or a clear upward trend; or by improving the model structure at the model layer, so that the model can adapt itself to the market conditions and make reasonable decisions. In addition, the simulation trades in this paper were only conducted in the Chinese market, which makes it challenging to execute short trades. In the future, we plan to conduct experiments in the U.S. and other markets in both long and short directions. Finally, time was the only criterion to test the effectiveness of the framework, and a real online rolling test will also be on the agenda.

## Figures and Tables

**Figure 1 entropy-25-01500-f001:**
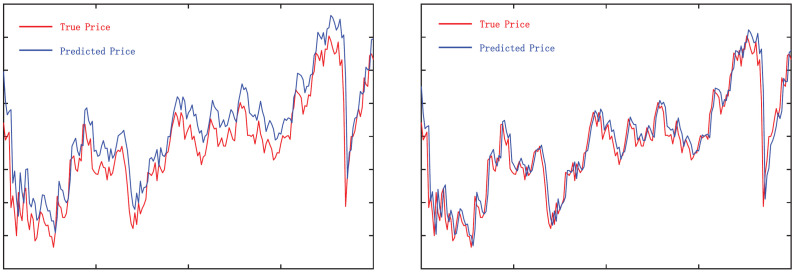
Example of stock price forecast.

**Figure 2 entropy-25-01500-f002:**
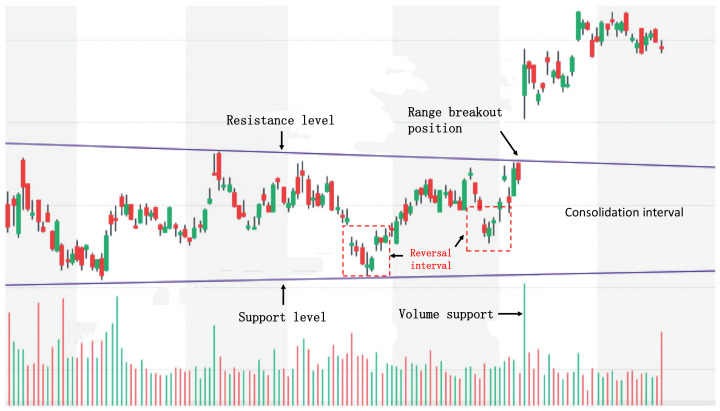
Example of Trading Range Breakout (TRB).

**Figure 3 entropy-25-01500-f003:**
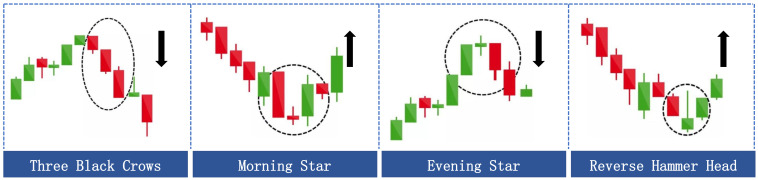
Example of four classic candlestick reversal patterns.

**Figure 4 entropy-25-01500-f004:**
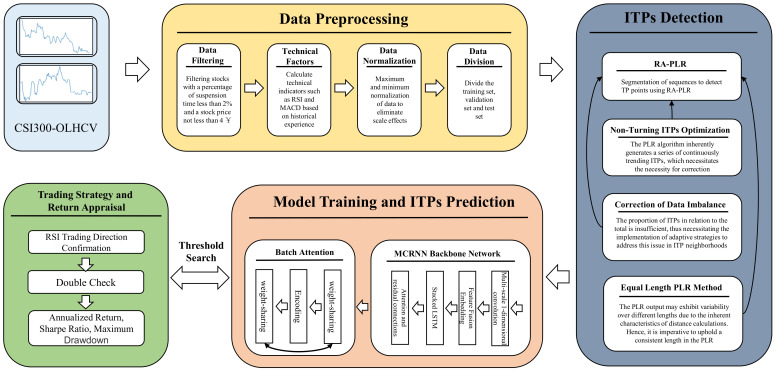
Architectural diagram of the ITP prediction framework.

**Figure 5 entropy-25-01500-f005:**
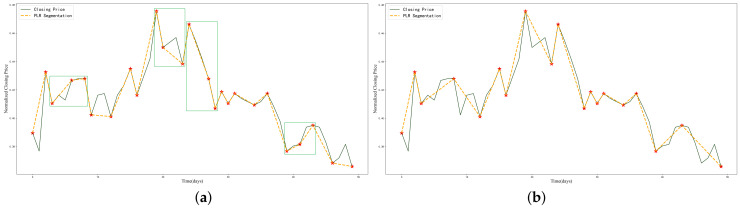
Example for optimization of non-turning ITPs: (**a**) before non-turning ITP optimization, (**b**) after non-turning ITP optimization.

**Figure 6 entropy-25-01500-f006:**
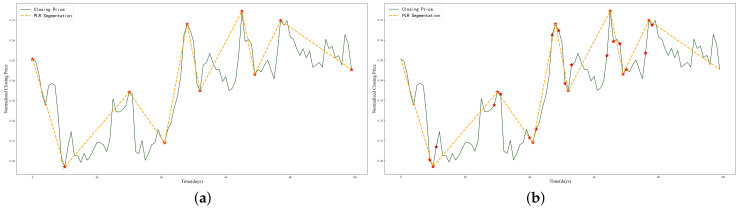
Example for ITP data augmentation: (**a**) before data enhancement, (**b**) after data enhancement.

**Figure 7 entropy-25-01500-f007:**
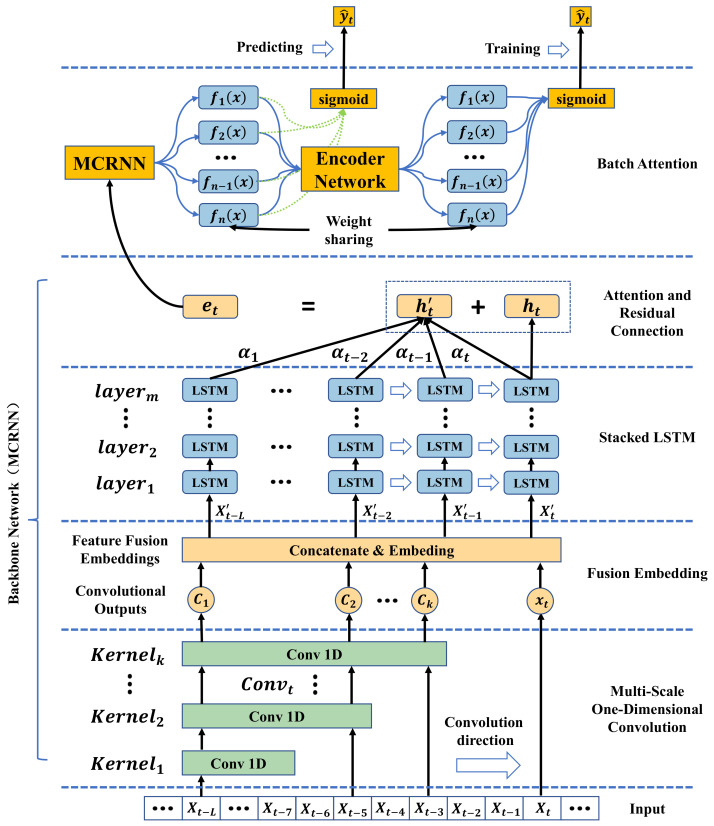
Structure of the proposed Multi-Scale Convolution Recurrent Neural Network (Batch-MCRNN) model for ITP prediction, with detailed layer connections indicated.

**Figure 8 entropy-25-01500-f008:**
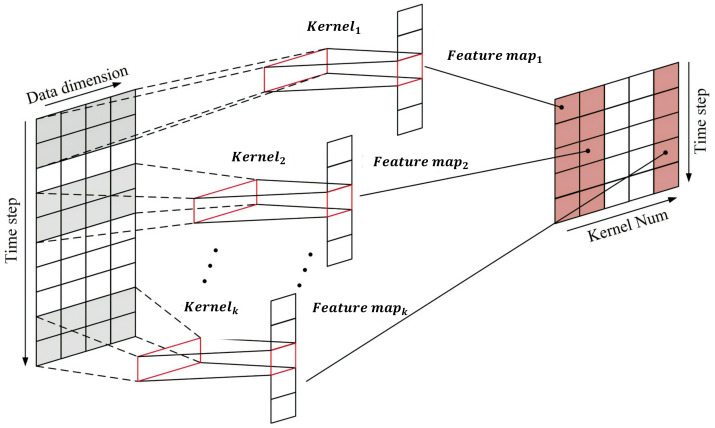
Diagram of one-dimension convolution at a single scale.

**Figure 9 entropy-25-01500-f009:**
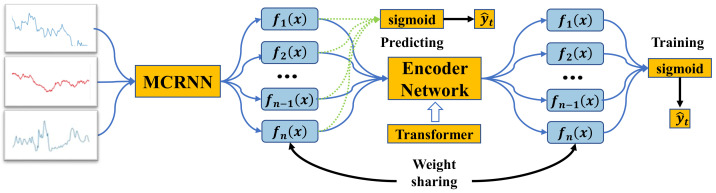
Structure of the batch attention mechanism.

**Figure 10 entropy-25-01500-f010:**
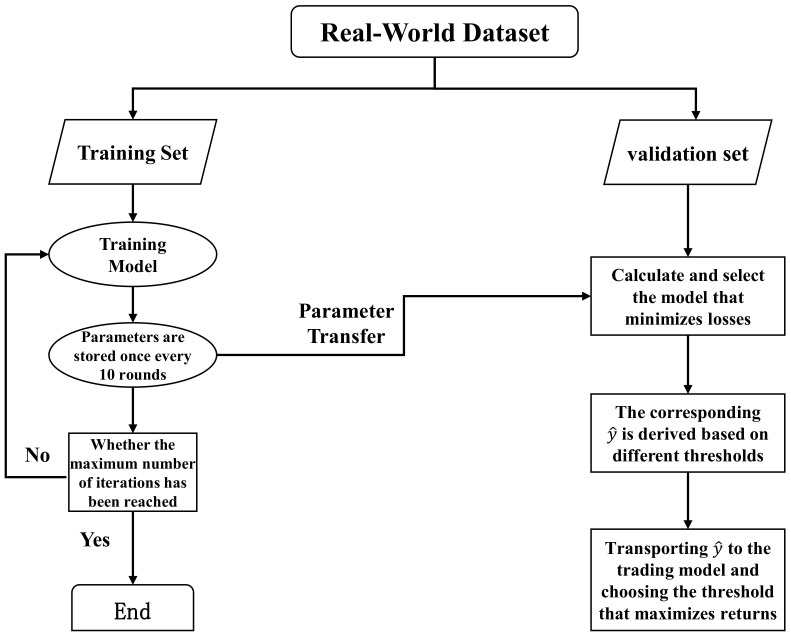
Flowchart of the returns retrospective threshold search mechanism.

**Figure 11 entropy-25-01500-f011:**
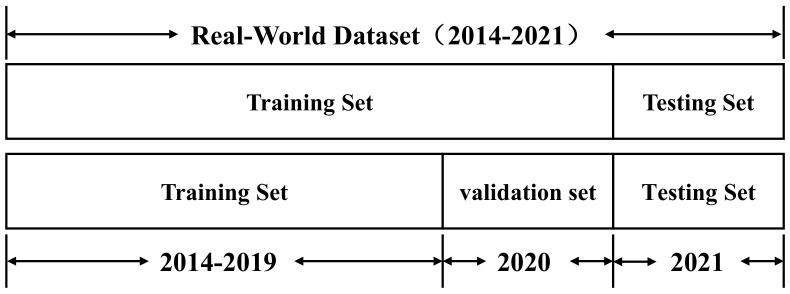
Diagram of the dataset division.

**Figure 12 entropy-25-01500-f012:**
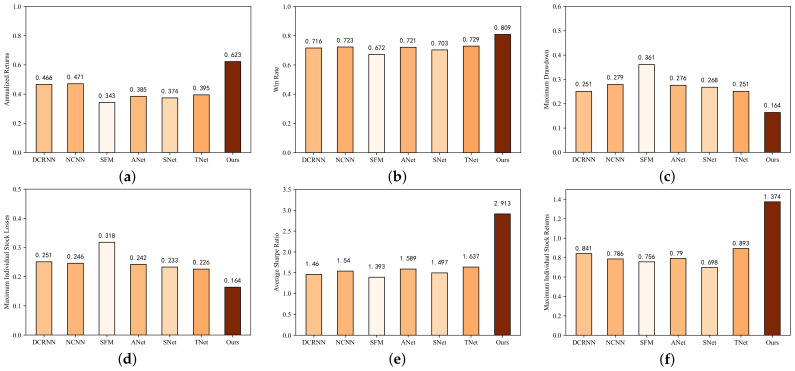
Comparison of the overall returns and risk control results obtained by each model using the same trading strategy on 100 stocks: (**a**) annualized returns, (**b**) win rate, (**c**) maximum drawdown, (**d**) maximum individual stock losses, (**e**) maximum individual stock returns, (**f**) average Sharpe ratio.

**Figure 13 entropy-25-01500-f013:**
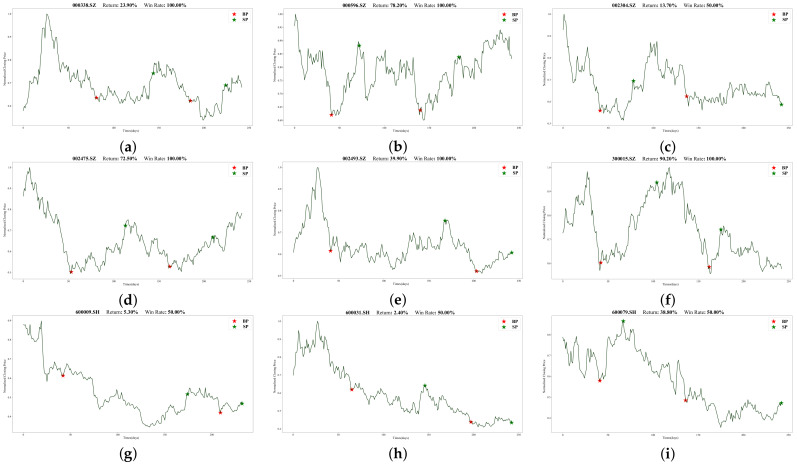
For each of the 18 stocks, the specific trading conditions are shown: (**a**) 000338.SZ, (**b**) 000596.SZ, (**c**) 002304.SZ, (**d**) 002475.SZ, (**e**) 002493.SZ, (**f**) 300015.SZ, (**g**) 600009.SH, (**h**) 600031.SH, (**i**) 600079.SH.

**Figure 14 entropy-25-01500-f014:**
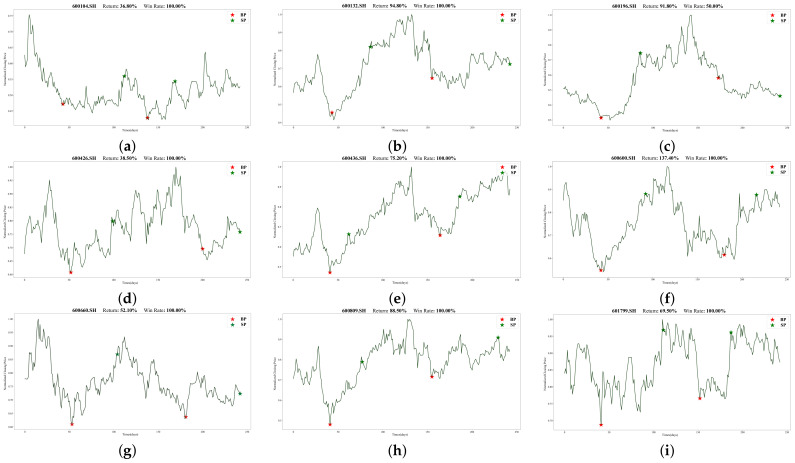
For each of the 18 stocks, the specific trading conditions are shown: (**a**) 600104.SH, (**b**) 600132.SH, (**c**) 600196.SH, (**d**) 600426.SH, (**e**) 600436.SH, (**f**) 600600.SH, (**g**) 600660.SH, (**h**) 600809.SH. (**i**) 601799.SH.

**Table 1 entropy-25-01500-t001:** Main technical indicators used in this work.

Abbreviation	Technical Factor	Formula
MA	Moving Average	$MA\left(N\right)=N^{-1}\Sigma_{i=1}^{N}S_{i,close}$
WMA	Weighted Moving Average	$WMA_{t}=\frac{\sum_{i=0}^{n-1}w_{i}Y_{t-i}}{\sum_{i=0}^{n-1}w_{i}}$
RSI	Relative Strength Index	$RSI_{n}=100-\frac{100}{1+EMA_{n_{up}}/EMA_{n_{\mathrm{down}\;}}}$
ALT	Amplitude of the Price Movement	$ALT_{i}\left(n\right)=\left(s_{i,\;\mathrm{high}\;}-s_{i,\;\mathrm{low}\;}\right)/s_{i,\;\mathrm{low}\;}$
ROC	Price Rate of Change	$ROC_{n}=\frac{s_{i,\;\mathrm{close}\;}-s_{i-n,\;\mathrm{close}\;}}{s_{i-n,\;\mathrm{close}\;}}$
ITK	Index for the Type of K-Line	$ITK_{i}=\left\{\begin{array}{cc} 1 & \;\mathrm{if}\;s_{i,\;\mathrm{close}\;}>s_{i,\;\mathrm{open}\;}\\ -1 & \;\mathrm{otherwise}\; \end{array}\right.$
MOM	Momentum	$MOM_{n}=\frac{s_{i,\;\mathrm{close}\;}}{s_{i-n,\;\mathrm{close}\;}}$
ADX	Average Directional Movement Index	$ADX_{n}=\frac{ADX_{n-1}*\left(N-1\right)+DX}{n}$

**Table 2 entropy-25-01500-t002:** Number of ITPs under different thresholds.

	δ=0.001	δ=0.005	δ=0.01	δ=0.02	δ=0.04
Count	25	24	19	15	4
Proportion	50%	48%	38%	30%	8%

**Table 3 entropy-25-01500-t003:** AUC for 18 stocks.

Stock	Model						
**Code**	**MCRNN**	**NCNN**	**SFM**	**AdvNet**	**StockNet**	**TEANet**	**Ours**
002493.SZ	0.725	0.725	0.716	0.693	0.643	0.729	**0.754**
002475.SZ	0.584	0.602	0.575	0.611	0.542	0.601	**0.685**
601628.SH	0.705	0.716	0.660	0.665	0.637	0.709	**0.778**
601799.SH	0.601	0.587	0.544	0.604	0.606	0.597	**0.657**
601012.SH	0.646	0.612	0.599	0.587	0.452	0.642	**0.689**
300015.SZ	0.586	0.607	0.608	0.634	0.599	0.621	**0.702**
600660.SH	0.611	0.638	0.594	0.655	0.631	0.641	**0.721**
600600.SH	0.559	0.580	0.530	0.614	**0.719**	0.588	0.704
600132.SH	0.612	0.608	0.620	0.653	0.656	0.642	**0.734**
000596.SZ	0.572	0.605	0.555	0.625	0.595	0.605	**0.659**
600809.SH	0.677	0.654	0.630	0.673	0.687	0.676	**0.719**
002304.SZ	0.601	0.613	0.567	0.604	0.525	0.619	**0.680**
600426.SH	0.701	0.711	0.689	0.775	**0.823**	0.702	0.751
600196.SH	0.781	0.759	0.752	0.796	**0.876**	0.773	0.823
000338.SZ	0.692	0.655	0.648	0.683	0.598	0.690	**0.760**
600079.SH	0.689	0.689	0.694	0.618	0.476	0.715	**0.774**
600436.SH	0.615	0.637	0.573	0.683	0.499	0.631	**0.685**
600009.SH	0.722	0.723	0.713	0.687	0.630	0.736	**0.832**
Average	0.649	0.651	0.626	0.659	0.622	0.662	**0.728**
σ	0.062	0.053	0.064	0.055	0.107	0.054	**0.050**

All model results were averaged over 5 training sessions.

**Table 4 entropy-25-01500-t004:** F1 score for 18 stocks.

Stock	Model						
**Code**	**MCRNN**	**NCNN**	**SFM**	**AdvNet**	**StockNet**	**TEANet**	**Ours**
002493.SZ	0.615	0.603	0.607	0.523	0.607	**0.642**	0.636
002475.SZ	0.318	0.335	0.385	0.475	0.416	0.433	**0.495**
601628.SH	0.494	0.504	0.498	0.553	0.438	0.530	**0.608**
601799.SH	0.605	0.617	0.601	0.509	0.648	0.616	**0.676**
601012.SH	0.626	0.639	0.594	0.485	0.480	0.629	**0.665**
300015.SZ	0.453	0.475	0.445	0.489	**0.590**	0.586	0.576
600660.SH	0.508	0.486	0.501	0.478	0.337	0.525	**0.586**
600600.SH	0.479	0.473	0.429	0.536	0.300	0.475	**0.555**
600132.SH	0.604	0.604	0.581	0.521	0.570	0.605	**0.661**
000596.SZ	0.458	0.494	0.464	0.498	0.600	0.497	**0.607**
600809.SH	0.492	0.474	0.483	0.482	0.442	0.502	**0.510**
002304.SZ	0.441	0.432	0.375	0.465	0.426	0.435	**0.474**
600426.SH	0.576	0.596	0.551	0.585	0.640	0.592	**0.652**
600196.SH	0.516	0.515	0.534	0.420	0.560	0.510	**0.566**
000338.SZ	0.529	0.564	0.525	**0.622**	0.620	0.563	0.601
600079.SH	0.493	0.495	0.452	0.510	0.502	0.505	**0.584**
600436.SH	0.461	0.457	0.408	0.494	0.334	0.460	**0.530**
600009.SH	0.371	0.375	0.550	0.479	0.562	**0.538**	0.520
Average	0.502	0.508	0.499	0.507	0.504	0.536	**0.583**
σ	0.081	0.081	0.072	**0.045**	0.108	0.063	0.059

All model results were averaged over 5 training sessions.

**Table 5 entropy-25-01500-t005:** MCC for 18 stocks.

Stock	Model						
**Code**	**MCRNN**	**NCNN**	**SFM**	**AdvNet**	**StockNet**	**TEANet**	**Ours**
002493.SZ	0.425	0.433	0.400	0.324	0.368	**0.474**	0.465
002475.SZ	0.158	0.163	0.174	0.277	0.093	0.234	**0.281**
601628.SH	0.216	0.223	0.192	0.251	0.260	0.271	**0.316**
601799.SH	0.157	0.150	0.159	0.214	0.162	0.215	**0.260**
601012.SH	0.231	0.227	0.201	0.210	0.174	**0.264**	0.209
300015.SZ	0.215	0.203	0.188	0.251	0.283	0.248	**0.331**
600660.SH	0.221	0.232	0.221	0.216	0.226	0.274	**0.304**
600600.SH	0.104	0.095	0.089	0.199	**0.259**	0.151	0.258
600132.SH	0.238	0.201	0.188	0.219	0.220	0.272	**0.290**
000596.SZ	0.054	0.050	0.028	0.174	0.110	0.128	**0.205**
600809.SH	0.247	0.245	0.212	0.205	0.250	**0.294**	0.221
002304.SZ	0.133	0.153	0.137	0.175	0.072	0.199	**0.225**
600426.SH	0.221	0.246	0.213	0.266	0.290	0.292	**0.326**
600196.SH	0.331	0.360	0.338	0.354	0.372	0.364	**0.394**
000338.SZ	0.276	0.267	0.234	0.256	0.157	0.308	**0.368**
600079.SH	0.310	0.325	0.327	0.333	0.389	0.389	**0.404**
600436.SH	0.161	0.184	0.167	0.276	0.184	0.229	**0.277**
600009.SH	0.238	0.270	0.246	0.243	0.116	0.303	**0.405**
Average	0.219	0.224	0.206	0.247	0.221	0.273	**0.308**
σ	0.084	0.089	0.084	**0.050**	0.094	0.080	0.073

All model results were averaged over 5 training sessions.

**Table 6 entropy-25-01500-t006:** Evaluation of the effectiveness of the DC method under Batch-MCRNN.

Strategy	Evaluation Metrics					
	Win Rate	Annualized Returns	Maximum Drawdown	Maximum Individual Stock Returns	Maximum Individual Stock Losses	Average Sharpe Ratio
DC	0.809	0.623	0.164	1.374	0.164	2.913
W/O DC	0.747	0.526	0.198	0.885	0.164	2.051

W/O DC indicates without the DC method.

## Data Availability

Publicly available datasets were analyzed for this study. All data can be found at the following: https://www.tushare.pro/, accessed on 1 October 2022.

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
