# Peer review of "A Framework for Enhancing Stock Investment Performance by Predicting Important Trading Points with Return-Adaptive Piecewise Linear Representation and Batch Attention Multi-Scale Convolutional Recurrent Neural Network"

_entropy, 2023, doi:10.3390/e25111500_

Round 1

Reviewer 1 Report

Comments and Suggestions for Authors

The authors present a comprehensive methodology for predicting Important Trading Points (ITPs) to bolster stock investment returns. They systematically address three pivotal elements: the identification of past ITPs, the forecasting of forthcoming ITPs, and the design of tailored trading strategies. The authors introduce a distinctive method, termed Return-Adaptive Piecewise Linear Representation (RA-PLR), to capture past ITPs, focusing on salient trading junctures that resonate with investor insights. For the anticipation of imminent ITPs, they put forth the Batch Attention Multi-Scale Convolution Recurrent Neural Network (Batch-MCRNN), adeptly harnessing the capabilities of convolutional neural networks, recurrent neural networks, and attention modalities. A Double Check (DC) approach further refines the trading schema, mitigating potential pitfalls by sidelining predictions perceived as tentative or fraught with elevated risk. Through thorough experimental scrutiny, the authors underscore the superiority of their paradigm over existing benchmarks, showcasing enhanced prediction precision, yield metrics, and risk modulation. I have the following concerns:

Major comments:

1. In the Introduction section, the authors delineate the motivation, content, and principal contributions of this study.  It would be advantageous for the authors to revisit and elaborate on methodologies previously documented in the literature, especially the benchmark methods compared in the experimental section. Emphasizing the innovations of the current work in comparison to previous approaches will further underscore its unique contributions.

2. In the Introduction, there's noticeable redundancy in content. For instance, the complexity of the stock market and its susceptibility to internal and external factors are mentioned multiple times. Additionally, the emphasis on "proposing a framework" is reiterated often towards the end of the section. I'd suggest reorganizing these paragraphs to eliminate repetition and enhance clarity.

3. In the Methodology section, the authors provide a clear overview of the components and techniques used in the framework. However, I have a query concerning the Batch-MCRNN. After the Fusion Embedding Layer, why did the authors opt for an LSTM model to learn the long-term temporal dependencies? Notably, following the LSTM, there's an application of a temporal attention mechanism to concentrate on relevant time steps, coupled with the introduction of residual connections for swifter convergence and enhanced expressiveness. Why not consider replacing the original LSTM layer directly with multi-layered Transformer Encoders? Such a model could not only capture long-term dependencies but also incorporate multi-head self-attention mechanisms and residual connections.

4. Given that the model pertains to the stock market, as noted by the authors in the conclusion, it might be beneficial to consider employing a rolling window approach during training and testing.

5. Within the trading strategy, the authors should clarify the rationale behind choosing the thresholds of 40 and 60 or reference any literature that guided this decision. Additionally, while the DC method has been introduced, further elaboration on the selection and values of α and β is necessary.

6. In the strategy section, the paper details the calculation of metrics for individual stocks as well as an aggregate metric for all stocks. However, there is a lack of comprehensive discussion on risk management in trading. For instance, have stop-loss or take-profit measures been implemented?

Comments on the Quality of English Language

In general the quality of the English is good. 

Author Response

Dear Editors and Reviewers:

Thank you for your letter and for the reviewers’ comments concerning our manuscript entitled “A Framework for Enhancing Stock Investment Performance by Predicting Important Trading Points with Return-Adaptive Piecewise Linear Representation and Batch Attention Multi-Scale Convolutional Recurrent Neural Network” (ID: entropy-2632382). Those comments are all valuable and very helpful for revising and improving our paper, as well as the important guiding significance to our researches. We have studied comments carefully and have made correction which we hope meet with approval.

Responds to the reviewers' comments:

  1. In the Introduction section, the authors delineate the motivation, content, and principal contributions of this study. It would be advantageous for the authors to revisit and elaborate on methodologies previously documented in the literature, especially the benchmark methods compared in the experimental section. Emphasizing the innovations of the current work in comparison to previous approaches will further underscore its unique contributions.

The author’s answer: We appreciate it very much for this good suggestion, and we have done it according to your ideas. In the introduction section, we further elaborate on the motivation, content, and primary contributions of our research. Additionally, we reexamine and discuss the baseline methods mentioned in the experiments, emphasizing the relationship between our proposed Batch-MCRNN model for ITPs prediction and prior research. We believe these improvements will highlight the unique contributions of our framework based on Revenue-Adaptive Piecewise Linear Representation (RA-PLR) and Batch-MCRNN in enhancing stock investment returns. Thank you once again for your review. Specific modifications can be found in lines 39-49 and 135-138 of the manuscript.

  1. In the Introduction, there's noticeable redundancy in content. For instance, the complexity of the stock market and its susceptibility to internal and external factors are mentioned multiple times. Additionally, the emphasis on "proposing a framework" is reiterated often towards the end of the section. I'd suggest reorganizing these paragraphs to eliminate repetition and enhance clarity.

The author’s answer: Thank you for your valuable feedback. We appreciate your attention to detail and have carefully considered your suggestions. In response to the redundancy issue in the introduction, we have reorganized the paragraph to eliminate repetition and enhance clarity. Specifically, we have simplified the discussion of stock market complexity and susceptibility to internal and external factors, ensuring a more concise and focused introduction of our proposed framework. Additionally, we have modified the description at the end of the section to improve logical coherence. Your input is highly valuable in enhancing the quality of our paper. For specific modifications, please refer to lines 21-27, 61-74, and 164-169 in the paper.

  1. In the Methodology section, the authors provide a clear overview of the components and techniques used in the framework. However, I have a query concerning the Batch-MCRNN. After the Fusion Embedding Layer, why did the authors opt for an LSTM model to learn the long-term temporal dependencies? Notably, following the LSTM, there's an application of a temporal attention mechanism to concentrate on relevant time steps, coupled with the introduction of residual connections for swifter convergence and enhanced expressiveness. Why not consider replacing the original LSTM layer directly with multi-layered Transformer Encoders? Such a model could not only capture long-term dependencies but also incorporate multi-head self-attention mechanisms and residual connections.

The author’s answer: Thank you for your attention to the details of our methodology. We considered several factors regarding the choice of using an LSTM model to learn long-term temporal dependencies in Batch-MCRNN.

  • Transformer Overfitting: While multi-layer transformer encoders capture long-range dependencies, they are easily overfitted on small data sets. We tried using transformer as the primary structure in our initial experiments, but severe overfitting led to poor performance. Therefore, we abandoned this structure.
  • Short-Term Effectiveness of LSTM: LSTM is less efficient to train due to its structural feature of series connection, which results in its inability to process data in parallel. However, thanks to the serial nature of LSTM, it performs very well in short sequences, although it cannot preserve long-distance temporal information. Training efficiency is not a major issue in small datasets, so LSTM gains our favor through its stability and effectiveness..
  • Transformer Model Complexity: Transformers typically have larger model sizes and require more computational resources. We combine LSTM with a time-attentive mechanism to learn the time dependence in the data. In addition, we utilize CNNs to capture local fluctuations and introduce residual connections to improve model convergence speed and expressiveness. Smaller models and better results are achieved.

In summary, using an LSTM with time attention allows us to effectively model temporal dependencies while addressing the limitations of both transformers and LSTMs. The combination of these techniques contributes to improved performance in Batch-MCRNN.

  1. Given that the model pertains to the stock market, as noted by the authors in the conclusion, it might be beneficial to consider employing a rolling window approach during training and testing.

The author’s answer: Thank you for your valuable feedback. The rolling window prediction method is often considered adequate in time series forecasting. However, we opted for a sequential dataset split in this study for two reasons. Firstly, in traditional time series forecasting, data points closer to the prediction time have higher information content because they follow the Markov property of the sequence. However, in predicting important trading points, we aim to uncover intrinsic patterns in trading volume and price changes within the time interval before significant trading points. We focus more on a small number of potentially crucial points rather than temporal continuity. Therefore, the rolling window method is not particularly necessary in this article. Secondly, the rolling window method has a notable drawback. As the rolling granularity decreases, computational requirements grow exponentially. Due to computational limitations, we did not employ the rolling window method. In summary, we chose to split the dataset sequentially for various reasons. Thank you once again for your suggestion.

  1. Within the trading strategy, the authors should clarify the rationale behind choosing the thresholds of 40 and 60 or reference any literature that guided this decision. Additionally, while the DC method has been introduced, further elaboration on the selection and values of α and β is necessary.

The author’s answer: We appreciate your thorough examination of the details and your thoughtful comments. Regarding the issue of selecting relative strength index (RSI) thresholds in our trading strategy, We apologize for not providing a clear rationale in the original manuscript. Due to the inherent predictive errors of the model, both overly high and overly low RSI thresholds can lead to a reduction in trading opportunities. Therefore, analyzing stock data with relatively balanced upward and downward trends, we observed that RSI values exhibit a bimodal distribution, with two peaks occurring around 60 and 40. Consequently, we optimized the RSI thresholds for trading direction to be 60 and 40. As for the Double Check (DC) method, we will provide detailed explanations in the revised manuscript regarding the selection and values of α and β to enhance our clarity. Specifically, we employed a grid search approach to determine the optimal α and β values that maximize returns on the validation set using specific models and thresholds. In our experiments, both values were set to 0.9. Once again, thank you for your insightful feedback, and we will ensure a thorough resolution of these issues. For specific modifications, please refer to the paper's lines 601-607 and 625-628.

  1. In the strategy section, the paper details the calculation of metrics for individual stocks as well as an aggregate metric for all stocks. However, there is a lack of comprehensive discussion on risk management in trading. For instance, have stop-loss or take-profit measures been implemented?

The author’s answer: Thank you for your interest in the strategy section of our paper. Based on your suggestions, we have made adjustments to the discussion regarding risk management. Specifically, concerning the issue of investment diversity, our simulation method involves simultaneous trading of 100 different stocks, significantly diversifying the trading risk. In terms of risk-reward, we conducted a total of 89 trades (buying and selling) within a 1-year trading cycle, with a win rate exceeding 80%. This indicates that there is a high likelihood of profit in each trade. The overall model’s Maximum Drawdown is 16.4%, and the Average Sharpe Ratio is 2.913 (calculated only for the 18 stocks traded more than once), both of which outperform typical scenarios. Our strategy demonstrates reasonable risk control. Regarding take-profit and stop-loss, due to the relatively low number of trades and individual stock losses not exceeding 20%, we did not employ additional stop-loss strategies to avoid reducing overall profitability. However, our prediction method focuses on significant trading points and incorporates dual verification strategies, implicitly accounting for some profit-taking and stop-loss capabilities. Finally, it’s important to note that our strategy is fully automated, eliminating any interference from human emotions. In summary, the trading strategy based on Batch-MCRNN exhibits sound risk management capabilities. For specific modifications, please refer to lines 653-663 in the paper.

Reviewer 2 Report

Comments and Suggestions for Authors

This is an interesting and well-written paper. The authors propose a new method to detect the Important Trading Points (ITPs) and then show the improvement in the stock return prediction by incorporating the detected ITPS. Here are my comments in no particular order.

1. This referee cannot understand the economic intuition well. What causes the ITPs? Why the variables the authors use can capture this nonlinearity arising from the ITPs? Why only use the historical price and trade volume but not control other important asset pricing factors and macro variables that are widely used in the literature? Are there any policy suggestions based on the results? Also, the authors do not provide enough information about the data they use. For example, are the used variables stationary or not? The current paper reads more like a technical report for a Quant company instead of an academic paper.

2. For the empirical evaluation, this referee is unsure how the authors determine the training set and testing set window. The authors also fail to provide the out-of-sample performance test. Hence, we do not know if the superiority is statistically significant or not. Also, despite the authors providing the results for each individual stock, it would be much more interesting to see the CSI 300 Index as a whole. Alternatively, it would be interesting for practitioners that the authors can argue their proposed model performs best in which industry or which scenario.

3. There are a few typos in the paper. The reference list is not well-ordered. The paper needs to be proofread carefully.

Author Response

Dear Editors and Reviewers:

Thank you for your letter and for the reviewers’ comments concerning our manuscript entitled “A Framework for Enhancing Stock Investment Performance by Predicting Important Trading Points with Return-Adaptive Piecewise Linear Representation and Batch Attention Multi-Scale Convolutional Recurrent Neural Network” (ID: entropy-2632382). Those comments are all valuable and very helpful for revising and improving our paper, as well as the important guiding significance to our researches. We have studied comments carefully and have made correction which we hope meet with approval.

Responds to the reviewers' comments:

  1. This referee cannot understand the economic intuition well. What causes the ITPs? Why the variables the authors use can capture this nonlinearity arising from the ITPs? Why only use the historical price and trade volume but not control other important asset pricing factors and macro variables that are widely used in the literature? Are there any policy suggestions based on the results? Also, the authors do not provide enough information about the data they use. For example, are the used variables stationary or not? The current paper reads more like a technical report for a Quant company instead of an academic paper.

The author’s answer: Thank you for your valuable feedback. I appreciate your thoughtful comments. Let me address your concerns:

  • Economic Intuition and ITPs: The Important Trading Points in our framework are detected using the Return-Adaptive Piecewise Linear Representation (RA-PLR) method. These ITPs represent critical moments in the stock market where significant price changes occur. The goal of this paper is to hopefully find the intrinsic patterns that lead to the creation of ITPs.
  • On the nonlinearity of the used data: Changes in stock price and trading volume are the final performance results after all the information is pooled, so they are the most direct variables that reflect the state of the stock and contain enough information for prediction. Data nonlinearity is captured by the Batch-MCRNN model proposed in this paper. The significant advantage of deep learning models over traditional statistical models is the portrayal of nonlinearity, and Batch-MCRNN combines the advantages of various structures such as CNN, LSTM, and the attention mechanism and is capable of mining the intrinsic laws that cause ITPs from three dimensions: time, space, and samples. Mining and the final experiment also prove the model's predictive ability.
  • Variables and Asset Pricing Factors: We mentioned in the introduction of the paper that other asset pricing factors and macro variables play a crucial role in stock market dynamics. However, stock prices and trading volume already contain enough information, except that they are full of noise and randomness, and this paper hopes to do a better job of tapping into the underlying patterns in the raw data without introducing additional variables so that it may perform well in other scenarios as well. Of course, in future research, we will also consider integrating more variables and information to provide a more comprehensive analysis.
  • Policy Suggestions: We agree that policy implications are essential. Our current paper primarily focuses on the technical aspects of ITP prediction. In practice, our framework could inform trading strategies, risk management, and investment decisions. However, we will further elaborate on policy implications in future work.
  • Data Information: We apologize for not providing detailed information on the data used. The stock data is also non-stationary, full of noise and randomness. We also used a unit root test to demonstrate this but did not account for the test results in the paper as our model does not presuppose stationarity.

Thank you once again for your thoughtful review.

  1. For the empirical evaluation, this referee is unsure how the authors determine the training set and testing set window. The authors also fail to provide the out-of-sample performance test. Hence, we do not know if the superiority is statistically significant or not. Also, despite the authors providing the results for each individual stock, it would be much more interesting to see the CSI 300 Index as a whole. Alternatively, it would be interesting for practitioners that the authors can argue their proposed model performs best in which industry or which scenario.

The author’s answer: Thank you for your insightful comments and suggestions. We appreciate the time and effort you have put into reviewing our paper. Here are our responses to your comments:

  • Training and Testing Set Window: As shown in Figure 11 in the paper, we divided the dataset chronologically according to the usual guidelines for time series forecasting, where the data from 2014-2019 is the training set, the data from 2020 is the test set, and the data from 2021 is the validation set.
  • Out-of-Sample Performance Test: Since our dataset is a mixed dataset of 100 stocks, the dataset is divided in chronological order, where both validation and test sets are not involved in training the model. Therefore, the performance of the test set can be considered as out-of-sample performance. Due to the specificity of time series, for new stocks, we recommend fine-tuning the model using their historical data to improve the model's performance in future forecasting for that stock.
  • CSI 300 Index Analysis: That’s an excellent suggestion. The cumulative decline of the CSI300 index by more than 1,000 points, or 21.3%, over the 2021 cycle is further evidence of the validity of our proposed ITPs predictive trading framework. We have added this analytical note to the revised paper. See lines 637-642 of the paper for more details.
  • Industry or Scenario Specific Performance: We agree that seeing where our model performs best will be interesting. However, this paper has a core starting point of increasing investment diversity, hence the selection of representative CSI300 constituents. We will look further into your proposal in future studies.

We hope these responses address your concerns adequately. We look forward to further suggestions that will help improve our paper

  1. There are a few typos in the paper. The reference list is not well-ordered. The paper needs to be proofread carefully.

The author’s answer: Thank you for your valuable feedback on our paper. We apologize for any typos in the paper. We have carefully proofread the manuscript and corrected all identified errors. If you could kindly point out specific instances, we would be grateful. The references are listed in the order in which they appear in the text, not by last name, and we would appreciate it if you could point out specific problematic areas. We hope that these revisions address your concerns. Please feel free to provide any further feedback or suggestions.

Round 2

Reviewer 1 Report

Comments and Suggestions for Authors

In general I am happy with the revisions made by the author, and their response. I recommend acceptance of the paper after minor editorial revision. 

Comments on the Quality of English Language

The quality of the English writing is good. 

Reviewer 2 Report

Comments and Suggestions for Authors

Thanks and I enjoy reading the revised version.